# Impact of Body Composition Parameters on Lung Function in Athletes

**DOI:** 10.3390/nu14183844

**Published:** 2022-09-16

**Authors:** Klara Komici, Fabio D’Amico, Sofia Verderosa, Iacopo Piomboni, Carmine D’Addona, Vito Picerno, Antonio Bianco, Andrea Caiazzo, Leonardo Bencivenga, Giuseppe Rengo, Germano Guerra

**Affiliations:** 1Department of Medicine and Health Sciences, University of Molise, 86100 Campobasso, Italy; 2Exercise and Sports Medicine Unit, Antonio Cardarelli Hospital, 86100 Campobasso, Italy; 3Department of Advanced Biomedical Sciences, University of Naples Federico II, 80131 Naples, Italy; 4Gérontopôle de Toulouse, Institut du Vieillissement, CHU de Toulouse, 31000 Toulouse, France; 5Department of Translational Medical Sciences, University of Naples Federico II, 80131 Naples, Italy; 6Istituti Clinici Scientifici Maugeri SpA Società Benefit (ICS Maugeri SpA SB), 82037 Telese Terme, Italy

**Keywords:** muscle mass, fat-free mass, lung function, FEV1, FVC, athletes

## Abstract

Background: Given the potential risk of unhealthy weight management, the monitoring of body composition in athletes is advised. However, limited data reveal how body composition measurements can benefit athlete health and, in particular, respiratory function. The aim of this study is to evaluate the impact of body composition on pulmonary function in a population of adult athletes. Methods: Data from 435 competitive adult athletes regarding body compositions parameters and spirometry are retrospectively analyzed. Results: Our study population consists of 335 males and 100 female athletes. Muscle mass and fat-free mass are significantly and positively associated with forced expiratory volume in the first second (FEV1) and forced vital capacity (FVC) in the male and female population, while waist-to-height ratio is negatively associated with FEV1, FVC, and FEV1/FVC in the male population. In multivariable analysis, muscle mass and fat-free mass show significant association with FEV1 and FVC in both males and females (*p* < 0.05), and waist-to-height ratio is significantly and inversely associated with FEV1 and FVC in males (*p* < 0.05). Conclusions: Fat-free mass and muscle mass are positively and independently associated with FEV1 and FVC in athletes of both genders, and waist-to-height ratio is inversely associated with FEV1 and FVC only among male athletes. These findings suggest that body composition in athletes may be helpful in monitoring respiratory function.

## 1. Introduction

Body composition derives from the dynamic balance between fat-free mass, muscle mass, fat mass, and body fluids, and constitutes a primary concern in different sport disciplines, as weight modifications and higher adiposity may negatively impact physical performances [1]. A considerable number of indirect methods such as energy X-ray absorptiometry, magnetic resonance, or bioelectrical impedance analysis (BIA), were implemented over past years to assess body composition parameters [1]. Muscle mass positively contributes to physical performance strength, while fat mass increase may mechanically hinder sport activities and exert metabolic effects, such as thermoregulation [2,3]. Regular monitoring of body composition is advised by the International Olympic Committee Medical Commission [4]. Among the gold standard tools employed in clinical practice, BIA represents a safe, non-invasive, low-cost, replicable, and easy-to-use method, which allows valid and reliable estimation of body composition [3,5]. Notably, a strong association between central adiposity and pulmonary function has been described [6]. Increased body mass index (BMI) and fat-free mass are associated with a reduction in FVC and FEV1 [6]. In contrast, another study reveals that BMI and other factors, such as percentage body fat and waist circumference, show a non-linear U-shaped association with FEV1 and FVC [7]. Higher FEV1 and FVC are associated with an improvement in physical performance [8], however, the physical active population is under-represented in these studies, and most of them focus on older participants with different comorbidities, which makes their participation in competitive sports categories difficult. Moreover, limited data reveal how body composition measurements can influence athlete health, in particular respiratory function. Compared to age-matched reference values, athletes have higher lung function [9,10,11], and an improvement in physical performance requires adaptation of respiratory function [12]. However, recent studies demonstrate a high incidence rate of respiratory diseases among athletes [13]. Exploring the predictive role of body composition parameters on lung function may be helpful in monitoring both respiratory function and adaptation to enhanced training. Therefore, the aim of this study is to evaluate the impact of the components of body composition on pulmonary function in a population of adult athletes.

## 2. Materials and Methods

### 2.1. Study Population

Data derive from Exercise and Sports Medicine Unit “Antonio Cardarelli Hospital”, Department of Medicine and Health Sciences, University of Molise, Campobasso, Italy, which provides regular clinical evaluation for athletes. According to the Italian law n. 158 2012, athletes interested in participating in competitive sports as indicated by Italian National Sport Federation should undergo annual or biannual medical evaluation including spirometry, resting and stress testing electrocardiogram, blood pressure monitoring, and urine examination. For the amatorial, recreational, non-competitive level, resting electrocardiogram is mandatory. In this retrospective analysis, consecutive athletes undergoing medical check-ups from January 2019 to January 2020 were included. Athletes participating in this study were selected among those meeting the following inclusion criteria: (a) age ≥ 18 years, (b) taking no inhaled corticosteroids, beta-mimetics or respiratory medications, (c) taking no dietary supplements for improving muscle mass, and d) willingness to participate in this study. Anamnestic data regarding the presence of respiratory diseases, smoking, and other cardiovascular risk factors were also collected. Sport discipline for all athletes, and classification based on the type of exercise and level of intensity, were registered [14,15]. All procedures were approved by the Institutional Review Board of Department of Medicine and Health Sciences, University of Molise (protocol number 11/22), and conducted in accordance with the declaration of Helsinki for studies on humans. All participants provided informed written consent for anonymous data collection prior to the study.

### 2.2. Anthropometric and Body Composition Measurements

Height was measured with a stadiometer (Wunder SA. Bl. srl A 200) in standing upright position, bare feet as indicated by the Anthropometric standardization reference manual [16]. Body mass, fat-free mass, percentage of fat-free mass, muscle mass, percentage of muscle mass, fat mass, percentage of fat mass, and BMI were estimated via the electrical impedance body composition analyzer TANITA BC-420MA (Tanita Corporation, Tokyo, Japan), as indicated by the manufacture. Waist circumference was measured with a tape and reported to the nearest 0.1 cm. Waist circumference was measured by positioning a flexible anthropometric tape parallel to the floor, between the last rib and the upper edge of the iliac crest, at the end of normal expiration [16]. Waist circumference divided by height was used to calculate waist-to-height ratio. A body shape index (ABSI) was calculated as described by other studies [17].

### 2.3. Spirometry

Spirometry was performed in accordance with recommended standards [18], with the subjects seated, wearing a nose clip, on the same day immediately after anthropometric and body composition measurements. FVC, FEV1, and FEV1/FVC ratio were measured using a clinical spirometer (Sensormedics Viasys Carefusion Vmax Encore 22).

### 2.4. Statistical Analysis

Descriptive data are presented as mean and ±standard deviation (SD), or number and percentage. Student’s T test and chi-squared test were performed for comparison of characteristics between genders. Correlations between spirometry data and body composition parameters were evaluated by Pearson’s correlation coefficient, and association between body composition parameters and pulmonary function was evaluated by univariable and multivariable regression analyses. Variables that show at least moderate Pearson’s correlation coefficient or result in significant association with pulmonary measurements in the univariable linear regression analysis, and those clinically considered as relevant for the study outcomes, were tested with multivariable linear regression analysis. Separated regression analysis was performed for male and female population considering the established differences in body composition and lung function [19,20]. Multivariable regression analysis was adjusted for age, sport discipline, smoking, and BMI, which are identified as confounders from previous studies [7,21,22]. In addition, other models were employed for MM and FFM correlation with FEV1 and FVC, also adjusted for height and sport intensity. Pulmonary function variables were selected as dependent variables, and body composition as independent variables. The independent variables were standardized based on their means and SD. R2 was considered a measure of the goodness-of-fit, and its partition according to the Shapley–Owen decomposition, was performed to measure variable contribution, expressed as a percentage of the global coefficient [23,24]. The presence of multicollinearity between explanatory variables was assessed with the variance inflation factor (VIF). Values smaller than 4 have been suggested as the maximum acceptable VIF thresholds [24]. The statistical significance was *p* ≤ 0.05 and data were analyzed by STATA SE 16.1 (StataCorp LLC, College Station, TX, USA).

## 3. Results

### 3.1. Characteristics of Population

The study population in this study comprises 435 athletes, 335 males and 100 females. All participants are competitive athletes. A total of 43.7% of participants practice high-intensity and 56.3% moderate-intensity exercise. In males, 62.4% perform endurance sport, 9.8% power sport, and 27.7% mixed sport. In females, 69% perform endurance sport, 23% power sport, and 8% mixed sport. BMI is significantly higher among male athletes: 24.4 ± 2.5 kg/m^2^ vs. 22.5 ± 2.9 kg/m^2^ *p*-value = ≤0.0001. All anthropometric measures are significantly higher among male athletes. All BIA parameters are significantly different between genders. Significant differences between males and females result also from comparison of spirometry measurements: FEV1: 4.2 ± 0.7 L vs. 3.2 ± 0.5 L *p*-value ≤ 0.0001; FVC 4.9 ± 0.9 L vs. 3.6 ± 0.6 L *p*-value ≤ 0.0001; FEV1/FVC 84.1 ± 9.3% vs. 87.7 ± 8.4% *p*-value = 0.01. Demographic data, anthropometric, BIA, and spirometry measurements are summarized in Table 1. Appendix A summarizes all sports types in our population. Higher FEV1 and FVC values are present in the high-intensity sports compared to moderate-intensity (Appendix A).

### 3.2. Unadjusted Linear Regression and Correlations

Unadjusted linear regression analysis reveals a significant association of fat-free mass and muscle mass with FEV1 and FVC in male and female athletes (all *p*-values ≤ 0.0001). Fat mass and waist circumference are inversely associated with FEV1 and FEV1/FVC in males. In females, fat mass results are inversely associated with FVC (regression coefficient: −0.173, *p*-value = 0.008), while waist circumference does not show significant associations. Waist-to-height ratio shows significant association with all spirometry measurements in male athletes, but not in females. ABSI results are significantly associated only with FEV1 in males. Data regarding univariate regression analysis in male and female population are summarized in Table 2 and Table 3. Fat-free mass and muscle mass present a moderate and strong correlation with FEV1 and FVC, respectively, as demonstrated by Pearson’s correlation coefficients. Correlation evaluation between body composition parameters and spirometry data in the overall population are reported in Appendix A.

### 3.3. Multivariable Regression Analysis

In male participants, fat-free mass and muscle mass show a significant independent association with FEV1 and FVC (*p*-value ≤ 0.0001). Waist-to-height ratio are negatively associated with both FEV1 and FVC: regression coefficient: −0.16 *p*-value = 0.003 R^2^: 0.33; and regression coefficient: −0.19 *p*-value = 0.006, R^2^: 0.17. Variables tested for association with FEV1/FVC fail to show significant association. Importantly, muscle mass is the most relevant factor associated with FEV1 and FVC, as shown by the percent fraction of global R^2^, 44.4% and 67.7% (Table 4).

In female athletes, fat-free mass is significantly associated with FEV1 (*p*-value: 0.011), explaining 18.8% of global R^2^: 0.35. The strength of fat-free mass association with FVC is higher, as revealed by fractional R^2^ contribution of 55.2%. Muscle mass also shows a relevant association with both FEV1 (*p*-value = 0.002) and FVC (*p*-value ≤ 0.0001). The partial contribution to the global R^2^ for muscle mass is 25.7% for FEV1 association and 51.7% for FVC. Table 5 summarizes multivariable regression analysis in female athletes. VIF analysis for multicollinearity are all below the maximum acceptable level (Table 4 and Table 5). In another model, also adjusted for exercise intensity, the association of muscle mass with FEV1 and FVC is significant in both males and females. In this model, fat-free mass results are significantly associated with FEV1 in both genders (Appendix A). Regression analysis adjusted for height added to the other confounders was also performed. Muscle mass confirms the significant correlation for FEV1 and FVC in males, and in females MM is significantly correlated with FVC. (Appendix A).

## 4. Discussion

The most relevant findings of the present research are: fat-free mass and muscle mass are positively and independently associated with FEV1 and FVC in athletes of both genders, and waist-to-height ratio is inversely associated with FEV1 and FVC only among male athletes.

The positive association between fat-free mass, muscle mass, and respiratory parameters in our study is consistent with results from previous studies. Reduction in fat-free mass is associated with a decline in pulmonary function [25,26,27], and higher lean body mass with higher lung function [28,29]. Vigorous physical activity is positively associated with markers of muscle mass, and individuals that practice regular physical activity are characterized by higher levels of FEV1 and FVC [21,30]. Furthermore, it is suggested that chronic endurance physical activity leads to adaptive changes in respiratory function [31]. The relationship between lung function and muscle mass may be intermediated by physical activity, however, the association of muscle mass and fat-free mass remains stable after adjustment for various confounders such as sport discipline, exercise intensity, age, BMI, or height. Of interest, muscle mass demonstrates a strong and independent association with FEV1 and FVC, as indicated by the partial contribution to R^2^. Possible mechanisms are likely to play a role: (a) increased muscle mass secondary to exercise may reflect an increased muscle strength [32,33], (b) reduction in fat mass, and in particular reduction in central adiposity, may increase the functional respiratory residual capacity [32], and (c) amelioration of insulin sensitivity may lead to higher cardiorespiratory fitness and muscle strength [34].

In our study, anthropometric and body components measures are significantly different when comparing female to male athletes. It is well-described that females are characterized by higher fat mass percentage and different fat mass distribution [35]. It should be mentioned that previous studies do not describe gender-related BMI differences in healthy subjects [36]. In our study, female athletes present a lower BMI than male athletes. BMI does not distinguish body composition, and differences may be either due to increased fat mass but also to reduced muscle mass. Indeed, the muscle mass component is lower among the female population. Furthermore, other studies including adult and master athletes report significantly lower BMI values in female athletes compared to males [37,38,39]. In our population, we include adult athletes of different age groups and different sport disciplines. In addition, other factors such as diet, nutrition, and social elements were not explored. All these conditions may play a role in BMI difference between genders.

Waist circumference is negatively associated with FEV1 and FEV1/FVC in male athletes, but does not show significant association in females. A possible explanation may be related to fat distribution, which may be more pronounced in the hip region for females and abdominal region for males. Indeed, waist circumference is an indicator of abdominal fat accumulation, which reduces pulmonary compliance and resistance, and impairs diaphragm movements [40]. Furthermore, a systematic review and meta-analysis concludes that an inverse relationship between WC and pulmonary function is present mainly in men [41]. The association between waist circumference and respiratory function is not significant after adjustment for confounders. A previous study reports that waist circumference is negatively associated with lung function, however, the authors do not consider physical activity as a confounder [42].

Waist-to-height ratio shows a significant inverse association with FEV1 and FVC in male athletes. Consistent with previous studies, waist-to-height ratio is shown to have a strong association with lung function [43,44], and it has been suggested as a good indicator for maintaining a healthy weight [45]. In our population, female athletes presented lower stature and differences related to fat distribution, which may, in part, explain the lack of association of waist-to-hip ratio among female athletes. In addition, it has been suggested that anthropometric measures translate differently to functional performance when comparing men and women [46].

Identification of body composition parameters as predictors of respiratory function in athletes is of great importance. Approximately one in five athletes is affected by lower respiratory disfunction, with the highest prevalence observed in those participating in elite endurance, aquatic, and winter-based sporting disciplines. From a theorical point of view, high ventilatory rate and weather or environmental conditions could increase the risk of small airways damage [47]. In addition, lack of respiratory predictive values for an athlete population can lead to mis-diagnosis of respiratory impairment.

Furthermore, fat-free mass is strongly associated with powerlifting performance [48], and is identified as a beneficial indicator for screening prospective young athletes [49]. In professional soccer players, fat-free mass significantly increases at mid- and end-season [50], and elite soccer players are reported to increase body mass, with 60% directly attributable to muscle mass [51]. Different sports disciplines, such as long-distance running, swimming, and cycling, require enhanced ventilation for meeting the gas exchange demand of the exercise [52]. Therefore, muscle mass may provide important information regarding the performance of respiratory function.

It is reported that weight loss may attenuate FEV1 and FVC decline, suggesting that lifestyle changes such as diet or physical activity may play an important role in lung function [29]. However, high BMI values are not always explained by increased fat mass, particularly among athletes whose musculoskeletal mass may be dominant. Total body weight and BMI do not distinguish between fat and muscle mass, and uncontrolled weight loss may lead to a reduction in muscle mass. Furthermore, fat mass and muscle mass could have different effects on lung function [53,54]. Our findings suggest that improvement in respiratory function in athletes is related mainly to increased muscle mass, and future studies should explore the benefit of physical activity and nutritional interventions on the attenuation of respiratory decline in different clinical contexts.

The monitoring of muscle mass by BIA in different time-points, such as before starting a program of training, at mid- and end-season for disciplines that require high adaptation of respiratory function, or at high risk for exercise-induced respiratory impairment, may be useful. In addition, the impact of regional body composition on respiratory fitness should be further evaluated.

Study limitations: This is a single-center study, including a limited number of participants. The frequency of female athletes is lower compared to males. We do not provide estimations regarding weekly energy expenditure, however, sport intensity is classified, as described elsewhere [15]. Another limitation may be related to spirometry examination performed in sitting position, since differences regarding measurements in standing position may not be excluded.

## 5. Conclusions

Fat-free mass and muscle mass are positively and independently associated with FEV1 and FVC in athletes of both genders, and waist-to-height ratio is inversely associated with FEV1 and FVC only among male athletes. These findings further suggest that body composition in athletes may be useful in monitoring respiratory performance.

## Figures and Tables

**Table 1 nutrients-14-03844-t001:** Population characteristics.

Characteristics	Male (*n*= 335)	Female (*n* = 100)	*p*-Value
Age, mean SD	37.3 ± 16.9	34.8 ±14.4	0.181
Weight kg, mean SD	74.9 ± 8.7	59.4 ± 7.7	≤0.0001
Height m, mean SD	1.75 ± 0.07	1.63 ± 0.06	≤0.0001
BMI kg/m^2^, mean SD	24.4 ± 2.5	22.5 ± 2.9	≤0.0001
WC cm, mean SD	85.6 ± 7.6	74.6 ± 7.9	≤0.0001
WHR, mean SD	0.49 ± 0.05	0.46 ± 0.05	≤0.0001
ABSI, mean SD	0.077 ± 0.005	0.073 ± 0.005	≤0.0001
Smoking *n* (%)	35 (10.4)	18 (18)	0.054
DM *n* (%)	6 (1.8)	0	0.178
Dyslipidemia *n* (%)	32 (9.5)	12 (12)	0.455
Hypertension *n* (%)	24 (7.1)	3 (3)	0.160
Arrythmias *n* (%)	14 (4.2)	4 (4)	0.384
Familiarity CV *n* (%)	42 (12.5)	17 (17)	0.248
FFM kg, mean SD	59.9 ± 11.7	43.4 ± 6.0	≤0.0001
FFM %, mean SD	80.3 ± 14.3	73.6 ± 10.2	≤0.0001
FM kg, mean SD	13.4 ± 7.3	15.4 ± 6.1	0.01
FM %, mean SD	17.6 ± 8.4	25.3 ± 7.1	≤0.0001
MM kg, mean SD	58.5 ± 6.9	41.9 ± 4.5	≤0.0001
MM %, mean SD	78.7 ± 6.3	70.7 ± 7.4	≤0.0001
FEV1 L, mean SD	4.2 ± 0.7	3.2 ± 0.5	≤0.0001
FVC L, mean SD	4.9 ± 0.9	3.6 ± 0.6	≤0.0001
FEV1/FVC % mean SD	84.1 ± 9.3	87.7 ± 8.4	0.01
Power sport *n* (%)	33 (9.8)	23 (23)	0.001
Mixed sport *n* (%)	93 (27.7)	8 (8)	≤0.0001
Endurance sport *n* (%)	209 (62.4)	69 (69)	0.238

BMI: body mass index; WC: waist circumference; WHR: waist-to-height ratio; ABSI: a body shape index; DM: diabetes mellitus; CV: cardiovascular disease; FFM: fat-free mass; FM: fat mass; MM: muscle mass; FEV1: forced expiratory volume during 1st second; FVC: forced vital capacity.

**Table 2 nutrients-14-03844-t002:** Univariate regression analysis of FEV1, FVC, and FEV1/FVC in male athletes.

Variable	Coefficient	95% CI	*p*-Value	R^2^
FEV1				
BMI	−0.14	−0.23–−0.05	0.002	0.029
FFM	0.24	0.15–0.33	≤0.0001	0.08
FFM%	0.11	0.04–0.19	0.004	0.02
FM	−0.11	−0.19–−0.04	0.004	0.02
FM%	−0.19	−0.27–−0.11	≤0.0001	0.06
MM	0.5	0.40–0.60	≤0.0001	0.21
MM%	0.3	0.18–0.36	≤0.0001	0.09
WC	−0.11	−0.21–−0.02	0.019	0.02
WHR	−0.29	−0.37–−0.21	≤0.0001	0.13
ABSI	−0.12	−0.21–−0.04	0.003	0.03
FVC				
BMI	−0.08	−0.18–0.03	0.138	0.006
FFM	0.33	0.23–0.45	≤0.0001	0.11
FFM%	0.12	0.04–0.22	0.006	0.02
FM	−0.07	−0.17–0.17	0.111	0.008
FM%	−0.17	−0.27–−0.07	0.001	0.03
MM	0.7	0.56–0.79	≤0.0001	0.28
MM%	0.3	0.15–0.37	≤0.0001	0.06
WC	−0.001	−0.11–0.11	0.984	-
WHR	−0.25	−0.34–−0.14	≤0.0001	0.07
ABSI	−0.09	−0.19–−0.004	0.062	0.01
FEV1/FVC				
BMI	−1.273	−2.26–−0.27	0.012	0.018
FFM	−0.679	−1.675–0.316	0.180	0.0054
FFM%	−0.058	−1.058–0.941	0.908	0.0001
FM	−1.089	−2.083–−0.095	0.032	0.013
FM%	−0.98	−1.983–0.006	0.052	0.011
MM	−0.458	−1.455–0.538	0.366	0.0024
MM%	1.175	0.182–2.168	0.021	0.016
WC	−0.13	−0.25–0.015	0.027	0.018
WHR	−1.78	−2.766–−0.797	≤0.0001	0.036
ABSI	−0.891	−1.82–0.039	0.060	0.010

BMI: body mass index; WC: waist circumference; WHR: waist-to-height ratio; ABSI: a body shape index; FFM: fat-free mass; FM: fat mass; MM: muscle mass; FEV1: forced expiratory volume during 1 s; FVC: forced vital capacity.

**Table 3 nutrients-14-03844-t003:** Univariable regression analysis of FEV1, FVC, and FEV1/FVC in female athletes.

Variables	Coefficient	95% CI	*p*-Value	R^2^
FEV1				
BMI	−0.04	−0.14–0.006	0.460	0.006
FFM	0.52	0.30–0.70	≤0.0001	0.18
FFM%	0.16	0.013–0.031	0.034	0.04
FM	−0.02	−0.16–0.11	0.71	0.001
FM%	−0.11	−0.25–0.025	0.108	0.03
MM	0.49	0.26–0.71	≤0.0001	0.16
MM%	0.07	−0.04–0.19	0.187	0.017
WC	0.04	−0.09–0.16	0.573	0.003
WHR	−0.07	−0.18–0.04	0.186	0.017
ABSI	0.024	−0.09–0.138	0.684	0.002
FVC				
BMI	0.006	−0.10–0.11	0.900	0.0002
FFM	0.414	0.306–0.522	≤0.0001	0.31
FFM%	0.15	−0.0004–0.311	0.051	0.038
FM	−0.173	−0.300–−0.047	0.008	0.055
FM%	−0.09	−0.24–0.045	0.177	0.018
MM	0.383	0.271–0.494	≤0.0001	0.268
MM%	0.05	−0.06–0.169	0.374	0.008
WC	−0.116	−0.245–0.121	0.075	0.025
WHR	−0.045	−0.15–0.067	0.428	0.006
ABSI	−0.002	−0.11–−0.11	0.973	-
FEV1/FVC				
BMI	−1.39	−2.91–0.13	0.072	0.032
FFM	0.68	−2.89–4.26	0.706	0.0015
FFM%	1.42	−0.83–3.68	0.213	0.015
FM	−1.18	−3.12–0.755	0.229	0.014
FM%	−1.45	−3.51–0.60	0.163	0.019
MM	0.72	−2.82–4.26	0.687	0.0017
MM%	1.31	−0.35–2.97	0.121	0.024
WC	−0.46	−2.35–1.42	0.626	0.002
WHR	−0.89	−2.5–0.72	0.277	0.012
ABSI	0.91	−0.75–2.58	0.280	0.011

BMI: body mass index; WC: waist circumference; WHR: waist-to-height ratio; ABSI: a body shape index; FFM: fat free mass; FM: fat mass; MM: muscle mass; FEV1: forced expiratory volume during 1 s; FVC: forced vital capacity.

**Table 4 nutrients-14-03844-t004:** Multivariable regression analysis in male athletes.

Variable	Coefficient	*p*-Value	R^2^	R^2^(f)%	VIF	VIF Mean
FEV1						
FFM	0.17	≤0.0001	0.35	17.2	1.05	1.1
FFM%	0.02	0.52	0.31	-	-	-
FM	−0.001	0.97	0.31	-	-	-
FM%	−0.07	0.09	0.32	-	-	-
MM	0.44	≤0.0001	0.45	46.4	1.23	1.2
MM%	0.13	0.019	0.32	7.1	1.9	1.4
WC	−0.06	0.21	0.47	-	-	-
WHR	−0.16	0.003	0.33	9.1	2.1	1.5
ABSI	−0.02	0.60	0.31	-	-	-
FVC						
FFM	0.27	≤0.0001	0.22	39.3	1.05	1.1
FFM%	0.07	0.120	0.15	-	-	-
FM%	−0.11	0.07	0.15	-	-	-
MM	0.67	≤0.0001	0.38	67.7	1.23	1.2
MM%	0.22	≤0.0001	0.17	17.7	1.9	1.4
WHR	−0.19	0.006	0.17	15.2	2.1	1.48
FEV1/FVC						
FM	−0.12	0.83	0.20	-	-	-
MM%	−0.6	0.43	0.12	-	-	-
WC	−0.9	0.21	0.13	-		
WHR	0.13	0.85	0.13	-	-	-

FFM: fat-free mass; FM: fat mass; MM: muscle mass; FEV1: forced expiratory volume during 1 s; FVC: forced vital capacity; WC: waist circumference; WHR: waist-to-height ratio; ABSI: a body shape index. This model was adjusted for age, smoking, and endurance sports discipline.

**Table 5 nutrients-14-03844-t005:** Multivariate analysis in female population.

Variable	Coefficient	*p*-Value	R^2^	R^2^(f)%	VIF	VIF Mean
FEV1						
FFM	0.31	0.011	0.35	18.8	1.39	1.24
FFM%	−0.03	0.759	0.31	-	-	-
MM	0.37	0.002	0.37	25.7	1.39	1.25
FVC						
FFM	0.52	≤0.0001	0.25	55.2	1.37	1.27
FM	0.059	0.660	0.15	-	-	-
MM	0.50	≤0.0001	0.26	51.7	1.38	1.29

FFM: fat-free mass; FM: fat mass; MM: muscle mass; FEV1: forced expiratory volume during 1 s; FVC: forced vital capacity. This model was adjusted for age, smoking, and endurance sports discipline.

## Data Availability

Data that underlie the results reported in this article will be available on request. Researchers should provide a methodologically sound proposal.

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
