# Peer review of "Impact of Body Composition Parameters on Lung Function in Athletes"

_nutrients, 2022, doi:10.3390/nu14183844_

Round 1

Reviewer 1 Report

The purpose of this retrospective study is to evaluate the impact of body composition on lung function in a population of adult athletes.

There is an extensive literature that has shown that a higher body fat percentage is associated with lower lung volumes. Direct and indirect measures of adiposity have similar associations with lung function and that adiposity has a greater effect on lung volumes in men than in women.

The authors conducted the retrospective study in a formally correct way. There are a few revisions to be made before possible publication.  

1. The difference in BMI between men and women is important. The BMI of the women in this study is much lower than that of the men. The authors should clarify why this difference also in view of the fact that the gender difference of %FM seems to be correct. 

2. The literature justifies the better lung function in athletes with less fat, particularly at the thoracic level. This simple difference, which can be assessed simply by BMi, could not be a quick explanation for the results of the paper. In summary, could the assessment of BMI not be sufficient? Could weight loss not be the cause of the improvement in lung function? 

3. Which sports do the assessed athletes play? At what intensity? Are there differences between the sports and the weekly METS with lung function?  

4.Correlation evaluation for statistically significant associations could improve the paper. Did the authors make a static correlation? 

5. The discussion is not very readable, also due to the presence of a very high number of abbreviations. The authors should make the discussion less confusing by emphasising the innovative aspects of the paper in relation to the relevant literature. 

Author Response

Reviewer#1

The purpose of this retrospective study is to evaluate the impact of body composition on lung function in a population of adult athletes.

There is an extensive literature that has shown that a higher body fat percentage is associated with lower lung volumes. Direct and indirect measures of adiposity have similar associations with lung function and that adiposity has a greater effect on lung volumes in men than in women.

The authors conducted the retrospective study in a formally correct way. There are a few revisions to be made before possible publication. 

  1. The difference in BMI between men and women is important. The BMI of the women in this study is much lower than that of the men. The authors should clarify why this difference also in view of the fact that the gender difference of %FM seems to be correct.

REPLY: Thank you for the comments. Indeed, previous studies have not described gender related BMI differences in healthy subjects, for instance: He et al Age- and sex-related differences in body composition in healthy subjects aged 18 to 82 years. 2018. 

However, other studies including adult and master athletes in their population have reported significantly lower BMI values in female athletes compared to males: Climstein M et al  Cardiovascular risk profiles of world masters games participants. J Sports Med Phys Fitness. 2018;58(4):489-96; Walsh J, et al. Body mass index for athletes participating in swimming at the World Masters Games. J Sports Med Phys Fitness. 2013;53(2):162-8; Gervasi M et al Muscular viscoelastic characteristics of athletes participating in the European Master Indoor Athletics Championship. Eur J Appl Physiol. 2017;117(8):1739-46.

 In our population we included adult athletes of different age groups and different disciplines , and this may be a possible explanation for BMI difference between genders.

We added a paragraph in discussion section where we clarified this difference. Please check page 9, lines 561-568.

  1. The literature justifies the better lung function in athletes with less fat, particularly at the thoracic level. This simple difference, which can be assessed simply by BMi, could not be a quick explanation for the results of the paper. In summary, could the assessment of BMI not be sufficient? Could weight loss not be the cause of the improvement in lung function?

REPLY:Thank you for the important suggestion. Total body weight and BMI do not distinguish between fat and muscle mass and uncontrolled weight loss may lead to reduction of muscle mass. Furthermore, fat mass and muscle mass could have different effects on lung function (Peralta GP, Fuertes E, Granell R, Mahmoud O, Roda C, Serra I, et al. Childhood Body Composition Trajectories and Adolescent Lung Function. Findings from the ALSPAC study. Am J Respir Crit Care Med. 2019;200(1):75-83; Park CH, Yi Y, Do JG, Lee YT, Yoon KJ. Relationship between skeletal muscle mass and lung function in Korean adults without clinically apparent lung disease. Medicine (Baltimore). 2018;97(37):e12281).Our findings suggest that improvement of respiratory function in athletes are related mainly to increased muscle mass. We added this paragraph in the discussion section. Please check page 10, lines 718-724.

  1. Which sports do the assessed athletes play? At what intensity? Are there differences between the sports and the weekly METS with lung function?

REPLY:In our study athletes practise different sports disciplines: swimming, cycling, football etc. Detailed information was added in supplementary materials table 1. In this retrospective analysis we did not estimate weekly energy expenditure, however sport intensity was classified as described by Mitchell JH, Haskell W, Snell P, Van Camp SP. Task Force 8: classification of sports. J Am Coll Cardiol. 2005;45(8):1364-7. This was added as a study limitation. Based on this information we performed other statistical analysis and higher FEV1 and FVC values were present in the high intensity sports compared to moderate intensity. Considering this important information, another multivariable regression analysis was performed adjusted also for sport intensity (moderate vs intense) and muscle-mass and fat-free resulted significantly associated to lung function parameters. This information was added in the result section and supplementary tables. (2-6) .

4.Correlation evaluation for statistically significant associations could improve the paper. Did the authors make a static correlation?

REPLY:Thank you for the suggestion. We performed a correlation test for body composition and respiratory parameters and added this information in result section. Fat-free mass and muscle mass presented a moderate and strong correlation coefficient with FEV1 and FVC. Please check also table 5, supp. material.

  1. The discussion is not very readable, also due to the presence of a very high number of abbreviations. The authors should make the discussion less confusing by emphasizing the innovative aspects of the paper in relation to the relevant literature.

REPLY:Following the Reviewer suggestions we modified the discussion section, discussed our results in relationship to other studies, emphasized our findings and in the revised version of our manuscript reported in words body composition parameters. Only in the tables they are presented in abbreviations. Please check discussion section.

We wish to thank the Reviewer for His/Her constructive criticism which helped us to improve our manuscript.

Reviewer 2 Report

Thank you for this interesting paper. Although the paper is interesting I have some concerns about the relevance of this study. Furthermore, the English writing should be improved and I would recommend to ask a native speaker to look at the language. Furthermore, the discussion must be revised with more structure. Below a point by point list of concerns which require attention.

11.  In the abstract there are some abbreviation which are not fully written for the first time. For instance FEV, FVC and WHR. Please carefully reread the abstract and add the needed information.

22.  In the introduction I miss the reason why this study is important. What kind of problem does knowing the correlation between lung function and body composition solve? Do many athletes have problems with lung functioning? And do you want to use this information for screening? Or is this information important to improve sport performance? Also in the discussion I miss a paragraph about the practical implications of these findings.

33.  In methods section the study population is described as athletes undergoing medical check-up. Can you please be more specific. At what level do this athletes participate are they at regional or national top of their sports? Or do also competitive athletes at lower recreational levels participate?

44.   In the statistical analysis it is described that a multivariate model is adjusted for age, sport discipline, smoking and BMI. Why did you chose these variables as confounders? are these variables shown to confound in earlier studies and is there a rationale for separated analysis for male and female athletes?

55.  Paragraph 3.1 is hard to read. I would suggest to rephrase some sentences.  For instance:

 “All participants were competitive athletes. In males 62% performed an endurance sport, 10% a power sport and 28% a mixed sport. In females….”

66.  In paragraph 3.2 line 138 you don’t have to explain that a univariate regression is not adjusted for confounders. Please remove from the sentence

77.  Paragraph 3.2 and 3.3 are difficult to read, because of the many reported results, please refer to your table and make a more readable paragraph, which supports your table by highlighting only the most important findings.

88.  First sentence of the discussion is too long and provides the same information twice. I would suggest to only use last part of the sentence. For instance: “FFM and MM are positively associated with FEV1 and FVC in male and female athletes, and WHR is inversely associated with FEV1 and FVC among male athletes.

99. The discussion is difficult to understand as it is not divided in paragraphs. Please rewrite with more structure to help the reader understand the main message. The discussion provides much information. However, it is unclear to me how this information is related to your observations.    

Author Response

Reviewer #2

Thank you for this interesting paper. Although the paper is interesting I have some concerns about the relevance of this study. Furthermore, the English writing should be improved and I would recommend to ask a native speaker to look at the language. Furthermore, the discussion must be revised with more structure. Below a point by point list of concerns which require attention.

  1. In the abstract there are some abbreviation which are not fully written for the first time. For instance FEV, FVC and WHR. Please carefully reread the abstract and add the needed information.

REPLY: Thank You for the comments. We revised the abstract and corrected abbreviations. In the revised version of our manuscript all body composition parameters are in words and respiratory parameters in abbreviations. Only in the tables are used abbreviations also for body composition parameters. We provided an extensive English language correction and sentences were rephrased.

  1. In the introduction I miss the reason why this study is important. What kind of problem does knowing the correlation between lung function and body composition solve? Do many athletes have problems with lung functioning? And do you want to use this information for screening? Or is this information important to improve sport performance? Also in the discussion I miss a paragraph about the practical implications of these findings.

REPLY: Thank You for the important remark and suggestions. Identification of body composition parameters as predictors of respiratory function in athletes is of importance because:

  1. a) approximately one in five athletes is affected by lower respiratory disfunction and lack of respiratory predictive values for athlete population can lead to mis-diagnosis of respiratory impairment.
  2. b) body composition elements have been identified as beneficial indicator for screening prospective young athletes
  3. c) different sports disciplines such as long-distance running, swimming and cycling require enhanced ventilation for meeting the gas exchange demand of the exercise. Therefor muscle mass may provide important information regarding monitoring of respiratory function and adaptation to exercise.

In addition, it has been reported that weight loss may attenuate FEV1 and FVC decline. However, total body weight and BMI do not distinguish between fat and muscle mass and uncontrolled weight loss may lead to reduction of muscle mass. Therefor our results may be helpful for future studies  exploring,  he benefit of physical activity and nutritional interventions on attenuation of respiratory decline in different clinical contexts.

In the introduction section we added why our study is important and also in discussion a paragraph about the clinical implications of our results with appropriate references. Please check page 2, lines 112-117, and page 10 and 11, 586-714.

  1. In methods section the study population is described as athletes undergoing medical check-up. Can you please be more specific. At what level do this athletes participate are they at regional or national top of their sports? Or do also competitive athletes at lower recreational levels participate?

REPLY:According to the Italian law n.158 2012, athletes interested in participating in competitive sports as indicated by Italian National Sport Federation should undergo annual or bi-annual medical evaluation including spirometry, resting and stress testing electrocardiogram, blood pressure monitoring and urine examination. In our population amatorial and recreational athletes were not included, since usually General Practitioners physicians perform the evaluation based on ECG and general medical examination (For their level it not mandatory spirometry). Our population included competitive athletes of different disciplines, regional and national level and moderate to intense exercise intensity. This information was added in the methods section, results and supp. materials table 1.  

  1. In the statistical analysis it is described that a multivariate model is adjusted for age, sport discipline, smoking and BMI. Why did you chose these variables as confounders? are these variables shown to confound in earlier studies and is there a rationale for separated analysis for male and female athletes?

REPLY:Separated regression analysis was performed for male and female population considering the established differences in body composition and lung function. Multivariable regression analysis was adjusted for age, sport discipline, smoking and BMI, which are identified as confounders from previous studies. In the revised version of our manuscript we added another model adjusted also for exercise intensity. Please check supplemental material table 6.

References:

-LoMauro A, Aliverti A. Sex differences in respiratory function. Breathe (Sheff). 2018;14(2):131-40.

-Bredella MA. Sex Differences in Body Composition. Adv Exp Med Biol. 2017;1043:9-27.

Cheng YJ, Macera CA, Addy CL, Sy FS, Wieland D, Blair SN. Effects of physical activity on exercise tests and respiratory function. Br J Sports Med. 2003;37(6):521-8.

-Lazovic-Popovic B, Zlatkovic-Svenda M, Durmic T, Djelic M, Djordjevic Saranovic S, Zugic V. Superior lung capacity in swimmers: Some questions, more answers! Rev Port Pneumol (2006). 2016;22(3):151-6.

  1. Paragraph 3.1 is hard to read. I would suggest to rephrase some sentences.For instance:

 “All participants were competitive athletes. In males 62% performed an endurance sport, 10% a power sport and 28% a mixed sport. In females….”

REPLY:Thank You! This was modified, please check paragraph 3.1

  1. In paragraph 3.2 line 138 you don’t have to explain that a univariate regression is not adjusted for confounders. Please remove from the sentence

REPLY: Thank You! This was removed.

  1. Paragraph 3.2 and 3.3 are difficult to read, because of the many reported results, please refer to your table and make a more readable paragraph, which supports your table by highlighting only the most important findings.

REPLY: Following your suggestion this was modified.

  1. First sentence of the discussion is too long and provides the same information twice. I would suggest to only use last part of the sentence. For instance: “FFM and MM are positively associated with FEV1 and FVC in male and female athletes, and WHR is inversely associated with FEV1 and FVC among male athletes.

REPLY:Following your suggestion this was modified.

  1. The discussion is difficult to understand as it is not divided in paragraphs. Please rewrite with more structure to help the reader understand the main message. The discussion provides much information. However, it is unclear to me how this information is related to your observations.

REPLY:Following the Reviewer suggestions we modified the discussion section, divided in paragraphs, discussed our results in relationship to other studies, emphasized our findings. Please check discussion section.

We wish to thank the Reviewer for His/Her constructive criticism which helped us to improve our manuscript.

Round 2

Reviewer 1 Report

The authors responded correctly to all my comments.

The paper is greatly improved.

Author Response

Reviewer #1

The authors responded correctly to all my comments.

The paper is greatly improved.

REPLY: Thank you for the comments and for appreciating our work.

Reviewer 2 Report

In my opinion, the paper is improved. However, I have some small concerns left before eventual publication.

1. results paragraph 3.2 starts with a result shown in the supplemental data. Please start with the main findings of your study and end the paragraph with the additional analysis.

2. in paragraph 3.2 sentences with referral to table 2 and 3 can be combined as they show the same results for male and female . Carefully reread your results try to provide a clear structure.

2. Paragraph 3 and 4 of the discussion line 234 to 250. Are highly suggestive. There is already much good information on both topics. However, I feel some dept in the discussion is missing, try to really explain why a difference in BMI is found in your study, or why no association was found between waist circumference and lung function in females. I feel the direction of the discussion is already there. However, some more elaboration is needed

3. Your conclusions state that your findings suggest that body composition in athletes may be useful in monitoring respiratory performance. Can you explain in your discussion how you should do this based on your results? Which parameter should be used and what type of action or research is needed to make this use clinical practice? In my opinion, this kind of recommendation can be made more specific than currently described.

Author Response

Reviewer #2

In my opinion, the paper is improved. However, I have some small concerns left before eventual publication.

  1. results paragraph 3.2 starts with a result shown in the supplemental data. Please start with the main findings of your study and end the paragraph with the additional analysis.

REPLY: Thank you for the comments and for appreciating our work.Following the reviewer suggestions, we started with the main findings related to body composition association with respiratory function and at the end replaced the Pearson's correlation analysis.

  1. in paragraph 3.2 sentences with referral to table 2 and 3 can be combined as they show the same results for male and female. Carefully reread your results try to provide a clear structure.

REPLY:Thank You for the Suggestions. We restructured this paragraph. In brief, we followed this structure: description of relationship between body composition components starting from muscle mass and fat free mass with respiratory parameters and anthropometric measures. We combined the description of data regarding males and females as indicated. Please check lines 158-172.  

  1. Paragraph 3 and 4 of the discussion line 234 to 250. Are highly suggestive. There is already much good information on both topics. However, I feel some dept in the discussion is missing, try to really explain why a difference in BMI is found in your study, or why no association was found between waist circumference and lung function in females. I feel the direction of the discussion is already there. However, some more elaboration is needed

REPLY: BMI does not distinguish elements of body composition. BMI differences may be related to higher fat mass but also to lower muscle mass. In addition, other factors as different sports discipline training, age groups, diet, nutritional and social may influence BMI differences related to gender. Regarding waist circumference and lung function a possible explanation may be related to fat distribution, which may be more pronounced in the hip region for females and abdominal region for males. Waist circumference is an indicator of abdominal fat accumulation which reduces pulmonary compliance and resistance and impairs diaphragm movements. These sentences were added in the text.

  1. Your conclusions state that your findings suggest that body composition in athletes may be useful in monitoring respiratory performance. Can you explain in your discussion how you should do this based on your results? Which parameter should be used and what type of action or research is needed to make this use clinical practice? In my opinion, this kind of recommendation can be made more specific than currently described.

REPLY: Monitoring of muscle mass by BIA in different time-points such as before starting a program of training, at mid and end-season for disciplines which require high adaptation of respiratory function or at high risk for exercise-induced respiratory impairment may be useful. In addition, the impact of regional body composition on respiratory fitness should be further evaluated.

We added this paragraph in discussion.

Once again, we wish to Thank the Reviewer for all his remarks and very helpful suggestions, which improved significantly the quality of our manuscript.
